# Synergistic activation of RARβ and RARγ nuclear receptors restores cell specialization during stem cell differentiation by hijacking RARα-controlled programs

Aysis Koshy[1],*, Elodie Mathieux[1],*, François Stüder[1], Aude Bramoulle[1], Michele Lieb[2], Bruno Maria Colombo[1], Hinrich Gronemeyer[2], Marco Antonio Mendoza-Parra[1]

How cells respond to different external cues to develop along defined cell lineages to form complex tissues is a major question in systems biology. Here, we investigated the potential of retinoic acid receptor (RAR)–selective synthetic agonists to activate the gene regulatory programs driving cell specialization during nervous tissue formation from embryonic carcinoma (P19) and mouse embryonic (E14) stem cells. Specifically, we found that the synergistic activation of the RARβ and RARγ by selective ligands (BMS641 or BMS961) induces cell maturation to specialized neuronal subtypes, and to astrocytes and oligodendrocyte precursors. Using RAR isotype knockout lines exposed to RAR-specific agonists, interrogated by global transcriptome landscaping and in silico modeling of transcription regulatory signal propagation, revealed major RARα-driven gene programs essential for optimal neuronal cell specialization and hijacked by the synergistic activation of the RARβ and RARγ receptors. Overall, this study provides a systems biology view of the gene programs accounting for the previously observed redundancy between RARs, paving the way toward their potential use for directing cell specialization during nervous tissue formation.

## Introduction

The potential of all-trans retinoid acid (ATRA) to induce differentiation of embryonic stem and embryonic carcinoma (EC) cells is well established (Soprano et al, 2007; Niederreither and Dollé, 2008). ATRA is a ligand for the three retinoic acid receptors (RARα, RARβ, and RARγ), and major medicinal chemistry efforts have resulted in the synthesis of ligands that are selective for each RAR isotype (de Lera et al, 2007; Álvarez et al, 2014). Multiple studies, including ours, demonstrated that P19 stem cells differentiate into neuronal precursors when treated with ATRA or the RARα-specific agonist BMS753, but they do not progress in differentiation when treated with the RARβ-specific agonist BMS641 or the RARγ-specific agonist BMS961 (Mendoza-Parra et al, 2016a).

Here, we have investigated the neuronal lineage–inducing potential of individual and combined subtype–specific retinoids in the two-dimensional monolayer culture of P19 EC and mouse embryonic stem cells (mESCs) (E14). We observe that, in addition to ATRA and RARα agonists, the combination of RARβ and RARγ agonists triggers a complex differentiation process generating a variety of neuronal subtypes, oligodendrocyte precursors and GFAP (+) astrocytes. This synergistic effect has been decorticated on the grounds of the RAR/RXR-driven gene programs, and the use of RAR subtype–deficient cells, which were instrumental for revealing the specificity of each of the synthetic ligands. Finally, we reveal that the RARβ+γ synergy, which involves a defined set of gene programs controlled by key master players, is antagonized in the presence of RARα, suggesting that an asynchronous activation of the various RARs leads to impaired neuronal specialization.

## Results

### Synergistic activation of RARγ and RARβ induces neuronal cell specialization in P19 embryonic stem cells

Using the well-established monolayer culture for efficient morphological P19 cell differentiation (Monzo et al, 2012; Mendoza-Parra et al, 2016a), we observed that after 10 d of treatment, ATRA or the RARα agonist BMS753 induced not only neuronal precursors, as revealed by immunofluorescence using the neuronal marker tubulin β-3 (TUBB3), but also mature neurons, as revealed by the microtubule-associated protein 2 (MAP2) (Fig 1A and B). In contrast, treatment with the RARβ-specific ligand BMS641 or the RARγ agonist BMS961 did not lead to neuronal differentiation. We only observed

---

[1]UMR 8030 Génomique Métabolique, Genoscope, Institut François Jacob, CEA, CNRS, University of Evry-val-d'Essonne, University Paris-Saclay, Évry, France    [2]Department of Functional Genomics and Cancer, Institut de Génétique et de Biologie Moléculaire et Cellulaire, Illkirch, France

Correspondence: mmendoza@genoscope.cns.fr
*Aysis Koshy and Elodie Mathieux contributed equally to this work

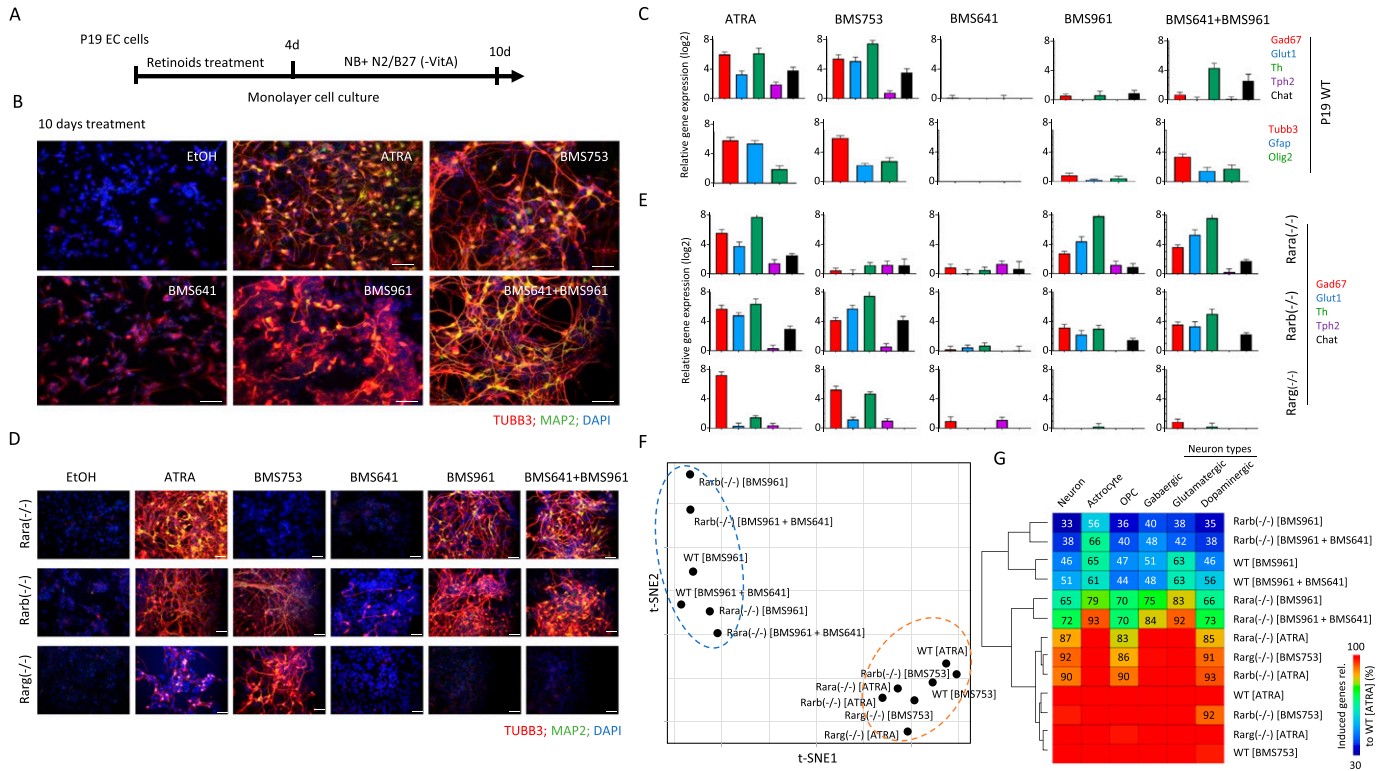

**Figure 1. Synergistic activation of the RARγ and RARβ induces neuronal cell specialization in P19 embryonic stem cells.**
**(A)** Schematic representation of the P19 cell differentiation assay. P19 cells cultured on monolayer are exposed to retinoids during 4 d to induce cell fate commitment; then, they are cultured for six more days on a synthetic medium (Neurobasal, NB) complemented with N2 and B27 (without vitamin A) supplements. **(B)** Immunofluorescence micrograph of WT P19 cells after 10 d of culture in presence of either ethanol (EtOH: vehicle control), all-trans retinoic acid, the RARα agonist BMS753, the RARβ agonist BSM641, the RARγ agonist BMS961, or the combination of RARβ and RARγ agonists. Cells were stained for the neuronal precursor marker TUBB3 (red) and the marker for mature neurons MAP2 (green). Nuclei were stained with DAPI (blue). **(C)** Top panel: RT–qPCR revealing the mRNA expression levels of gene markers associated with GABAergic (Gad67), glutamatergic (Glut1), dopaminergic (Th), or cholinergic (Chat) neuronal subtypes in samples treated with the indicated RAR agonists. Bottom panel: RT–qPCR mRNA gene expression levels of the glial fibrillary acidic protein (Gfap), the oligodendrocyte transcription factor 2 (Olig2) and the neuronal precursor marker Tubb3. **(D)** Immunofluorescence micrograph of P19 Rar-null mutant cells after 10 d of treatment with the aforementioned RAR agonists. **(E)** RT–qPCR mRNA expression levels of gene markers associated with the aforementioned neuronal subtypes assessed on P19 Rar-null mutant cells. **(F)** t-Distributed stochastic neighbor embedding analysis of differential gene expression readouts assessed on global transcriptomes performed on WT or Rar-null cells treated with specific agonists (10 d). Differential gene expression has been assessed relative to the ethanol-treated control sample (fold change levels >4). **(G)** Fraction of up-regulated genes (fold change levels >4) associated with markers corresponding to specialized cells relative to those observed on the gold-standard WT all-trans retinoic acid–treated sample. Fraction levels higher than 95% are only displayed with the heatmap color code (red).

neuronal-like cells presenting short neurite outgrowth structures, devoid of MAP2 immunostaining. Surprisingly, the combination of these two synthetic agonists (BMS641 and BMS961) restored neuronal differentiation presenting neurite outgrowth characteristics as similar as those observed on ATRA- or BMS753-treated samples (Fig 1B).

Neuronal maturation has been further supported by RT–qPCR assays revealing significant transcript levels associated with markers for GABAergic (*Gad67*), glutamatergic (*Glut1*), dopaminergic (*Th*), or cholinergic (*Chat*) neuronal subtypes in samples treated with ATRA and BMS753 (Fig 1C). Combined exposure to RARβ+γ agonists (BMS641 and BMS961) presented significant expression levels only for the markers *Th* and *Chat*, suggesting not only a partial neuronal subtype differentiation in comparison with ATRA or BMS753 treatment, but also the necessity of a more comprehensive strategy (global transcriptomes) to evaluate the cell specialization success. In addition to neuronal cell specialization, RT–qPCR assays also revealed significant expression levels of the glial fibrillary

acidic proteins (*Gfap*) and oligodendrocyte transcription factor 2 (*Olig2*) genes, indicative of the presence of astrocytes and oligodendrocyte precursors, both in ATRA and BMS753 treatment and in the combination of BMS641 and BMS961 agonists (Fig 1C).

Although the combined exposure to RARβ+γ agonists (BMS641 and BMS961) led to morphological neuronal cell specialization, the evaluated markers present systematic lower levels than those observed in ATRA or BMS753. As this could be due to a potential inhibitory effect of non-liganded RARα, we engineered P19 cells deficient for each of the RARs using the CRISPR/Cas9 technology. Surprisingly, the absence of the expression of either RARα, RARβ, or RARγ receptor directly affected the expression of the non-deleted RARα and the RARβ receptors, notably by preserving their induction after 96 h of treatment (Fig S1). Furthermore, P19 Rara(−/−) cells gave rise to mature neurons when treated not only with ATRA, but also with the RARγ agonist BMS961 or the combination of RARβ+γ ligands (BMS641 and BMS961), as revealed by TUBB3/MAP2 immunostaining (Fig 1D) and the high expression of transcripts

associated with neuronal subtypes *Gad67*, *Glut1*, and *Th* (Fig 1E), *Tubb3* and the oligodendrocyte marker *Olig2* (Fig S2). The observed enhanced neuronal differentiation in the presence of the RARγ agonist in P19 Rara(−/−) relative to WT cells is in agreement with previous studies on the functional redundancy of RAR subtypes during endodermal (F9) and neuronal (P19) cell differentiation (Roy et al, 1995; Taneja et al, 1995, 1996). Neuronal differentiation was also observed in RARβ-deficient cells treated with not only ATRA or the RARα agonist (BMS753), but also the RARγ agonist (BMS961) and the RARβ+γ agonist combination (BMS641 and BMS961) (Fig 1D and E). Finally, the Rarg(−/−) cells entered neuronal differentiation only in the presence of ATRA or BMS753, in agreement with our earlier finding that the RARα-dependent gene program directs the neuronal cell fate of P19 cells (Mendoza-Parra et al, 2016a).

Although neuronal differentiation performance driven by RARβ+γ agonist treatment on WT, and Rara- and Rarb-deficient cells was evaluated by immunostaining and RT–qPCR assays targeting few marker genes, we reasoned that a comprehensive strategy could reveal potential differences among these multiple conditions. Global transcriptome assays performed on WT or RAR subtype–deficient cells treated with specific agonists during 10 d revealed between 1,340 and 2,250 up-regulated genes (fold change levels >4 relative to the ethanol control) allowing to query for cell specialization signatures and their corresponding divergencies between samples (Fig S3). Indeed, a t-distributed stochastic neighbor embedding analysis of their differentially expressed genes revealed two major groups. The first comprises WT or RAR subtype–deficient cells treated with either ATRA or the RARα agonist (BMS753) (Fig 1F). The second group gathers samples treated with the RARγ (BMS961) or the combination of RARβ and RARγ agonists (BMS641 and BMS961). This second group displays significant disparities among their components, with the transcriptomes of RARβ+γ agonist–treated Rara(−/−) cells being closer to group 1 than the others, in line with the observed neuronal differentiation (Fig 1F).

To further understand this classification through better characterization of the cell specialization signature during these differentiation conditions, we have collected an ensemble of gene markers associated with neurons (1,352 genes), astrocytes (501 genes), and oligodendrocyte precursors (OPCs: 501 genes), and stratified on GABAergic (318 genes), glutamatergic (311 genes), and dopaminergic (513 genes) neuronal subtypes (Table S1) (Hook et al, 2018; Tasic et al, 2018; Voskuhl et al, 2019). By comparing the number of up-regulated genes in the WT ATRA treatment with these comprehensive lists of markers, we have revealed that ~30% of them are associated with markers corresponding to specialized cells (632 from 2,158 up-regulated genes from which 401 are associated with neurons, 100 with astrocytes, and 131 with oligodendrocyte precursors) (Fig S3). Considering the up-regulated genes associated with specialized cells in the WT ATRA condition as the gold standard for optimal cell differentiation, we have revealed that most of the ATRA- or BMS753-treated samples presented similar amounts of genes associated with specialized cells. Indeed, the Rara(−/−) mutant sample treated with ATRA recapitulated ~83% of the gold-standard up-regulated genes associated with oligodendrocyte precursors, ~87% for neuronal-associated gene markers, and ~85% for the dopaminergic neuronal subtype; a

similar behavior is observed for the Rarb(−/−) mutant treated with ATRA or the Rarg(−/−) mutant treated with the BMS753 ligand (Fig 1G). In contrast, samples treated with the RARγ agonist BMS961 give rise to cell marker levels of only ~30% in the context of the Rarb(−/−) mutant, and to ~70% in the context of the Rara(−/−). Importantly, treating the Rara(−/−) mutant with the combination of the RARβ+γ agonist (BMS641 and BMS961) leads to >70% of the gold-standard levels associated with neuronal cells, and even more than 90% for astrocyte or the glutamatergic neuronal cell type, demonstrating that the use of a synergistic RARβ+γ agonist treatment on RARα subtype–deficient cells leads to enhanced restoration of cell specialization during P19 stem cell differentiation.

## P19 differentiation driven by the combination of RARβ and RARγ agonists presents a delayed expression of cell specialization markers

To assess the temporal evolution of gene expression during RAR ligand–induced neuronal differentiation and to associate specific gene programs with the appearance of neuronal cell subtypes, we generated global transcriptomes after 2, 4, and 10 d of treatment of WT P19 cells. This has been performed for samples treated with either the pan-agonist ATRA—as a gold-standard treatment—the RARα-specific agonist BMS753, or the combination of RARβ and RARγ agonists (BMS641 and BMS961), shown to induce neuronal cell specialization.

Differential gene expression through the aforementioned timepoints —assessed during ATRA treatment— was classified into 14 relevant gene co-expression paths, defined herein as a group of genes with similar temporal changes of expression levels (Fig 2A). For instance, Path 1 (1,195 genes) corresponds to genes up-regulated (fold change >2 relative to vehicle at d0) after 2 d of treatment and remained overexpressed until day 10. Paths 2 (725 genes) and 4 (582 genes) comprise late up-regulated genes, induced only at d4 and d10, respectively (Fig 2A). A gene ontology term analysis performed over the first seven gene co-expression paths—associated with up-regulated genes at least at one timepoint—reveals that Path 1 comprises genes involved in neuronal differentiation, nervous system development, or axon guidance, and Path 2, in axonogenesis or axon development, whereas Path 4 is associated with chemical synaptic transmission, synaptic vesicle budding, or synapse organization, in agreement with the time of induction along the neuronal differentiation lineage and subtype specification (Fig 2B).

As expected, the up-regulated gene co-expression paths of WT P19 cells contained also markers indicative of the various specialized cells, as revealed by comparison with the aforementioned collection resource (Table S1). As illustrated in Fig 2C, 194 genes of the early-responsive gene co-expression Path 1 corresponded to neuronal markers, whereas 65 genes corresponded to astrocytes and other 68 genes corresponded to oligodendrocyte precursors (OPCs). The intermediate-responsive Path 2 presented 107 genes associated with neurons, 32 with astrocytes, and 35 with OPCs, whereas the late-responsive Path 4 presented 93 neuronal markers, 12 genes associated with astrocytes, and 30 others associated with OPCs. These kinetics indicate that neuronal differentiation precedes glial cell emergence, in agreement with previous findings in

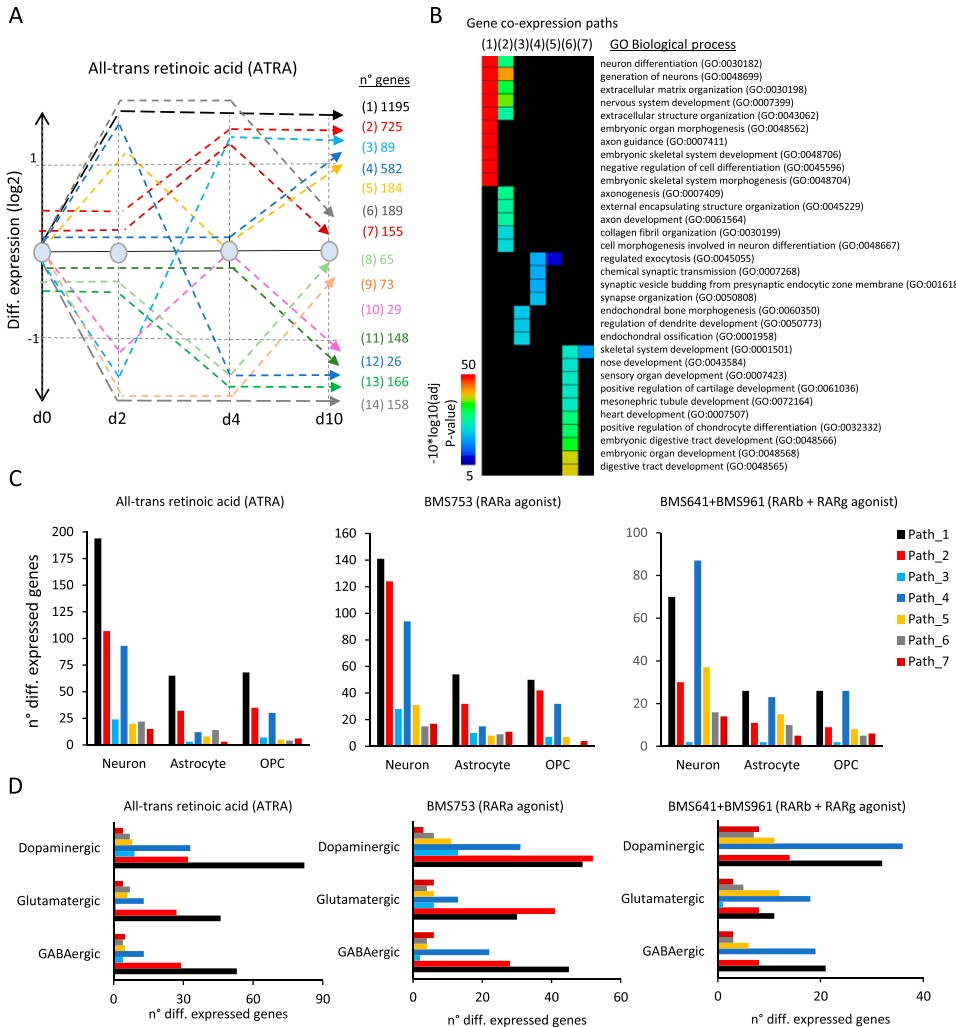

**Figure 2. Temporal gene co-expression analysis during cell specialization driven by retinoid treatment.**
**(A)** Stratification of the temporal transcriptome profiling during WT P19 cell differentiation driven by ATRA treatment. Transcriptomes were assessed on samples collected at 2, 4, and 10 d of treatment. Dashed lines correspond to groups of differentially co-expressed genes (gene co-expression paths; fold change levels >2). The numbers of genes composing each of the co-expression paths are displayed (right). **(B)** Gene ontology analysis on gene co-expression paths displayed in (A) associated with up-regulated events. **(C)** Number of genes per co-expression path corresponding to neuronal, astrocyte, or oligodendrocyte precursor cell types assessed during ATRA (left panel), BMS753 (middle panel), and BMS641 + BMS961 (right panel) treatment. **(D)** Similar to (C) but corresponding to dopaminergic, glutamatergic, and GABAergic neuronal subtypes.

in vivo and in vitro mammalian systems (reviewed in Miller & Gauthier [2007] and Hirabayashi & Gotoh [2010]). All other paths presented less than 25 genes corresponding to the aforementioned cells, indicating that early (Path 1)-, middle (Path 2)-, and late (Path 4)-responsive co-expression paths are the most relevant for describing cell specialization (Fig 2C and D).

Although WT P19 cells treated with the RARα agonist BMS753 presented relatively similar transcriptome kinetics, the number of markers associated with specialized cells retrieved on Path 1 was lower than that observed on the gold-standard ATRA treatment (141 genes associated with neurons, ~54 genes with astrocytes, and 50 genes with OPCs). This observation for Path 1 has been further enhanced on cells treated with RARβ and RARγ agonists (BMS641 and BMS961), including in addition a significant reduction in the number of gene markers associated with specialized cells on Path 2. In contrast, the number of gene markers observed on the late-responsive Path 4 remained rather unchanged (87 genes associated with neuronal markers, 23 with astrocytes, and 26 with OPCs) (Fig 2C). This observation suggests that although treatment with the combination of the RARβ+γ agonist gives rise to specialized cells, their differentiation process is delayed over time relative to that

observed under the ATRA treatment. This is also supported by the fact that cells under the RARβ+γ agonist treatment present gene markers associated with specialized neurons preferentially found on the late-responsive Path 4 (Fig 2D).

## Reconstruction of gene regulatory networks (GRNs) involved in cell specialization driven by retinoid treatment

Our previous work has shown that ligand binding of retinoid receptors triggers a cascade of events, which leads to the dynamic activation of other transcription factors (TFs), which then regulate their cognate targets. This cascade of transcription regulatory events can be reconstructed by integration of transcription factor–target gene (TF-TG) databases in the temporal transcriptome analysis (Cahan et al, 2014; Mendoza-Parra et al, 2016a). This way, GRNs can be reconstructed and master regulator genes deduced (Cholley et al, 2018).

Herein, we have reconstructed a master GRN from the integration of the temporal transcriptomes assessed on WT P19 cells treated with the pan-agonist ATRA, covering 10 d of cell treatment, with TF-TG annotations (CellNet database [Cahan et al, 2014]). This master GRN, composed of 1,156 nodes (genes) and 17,914 edges

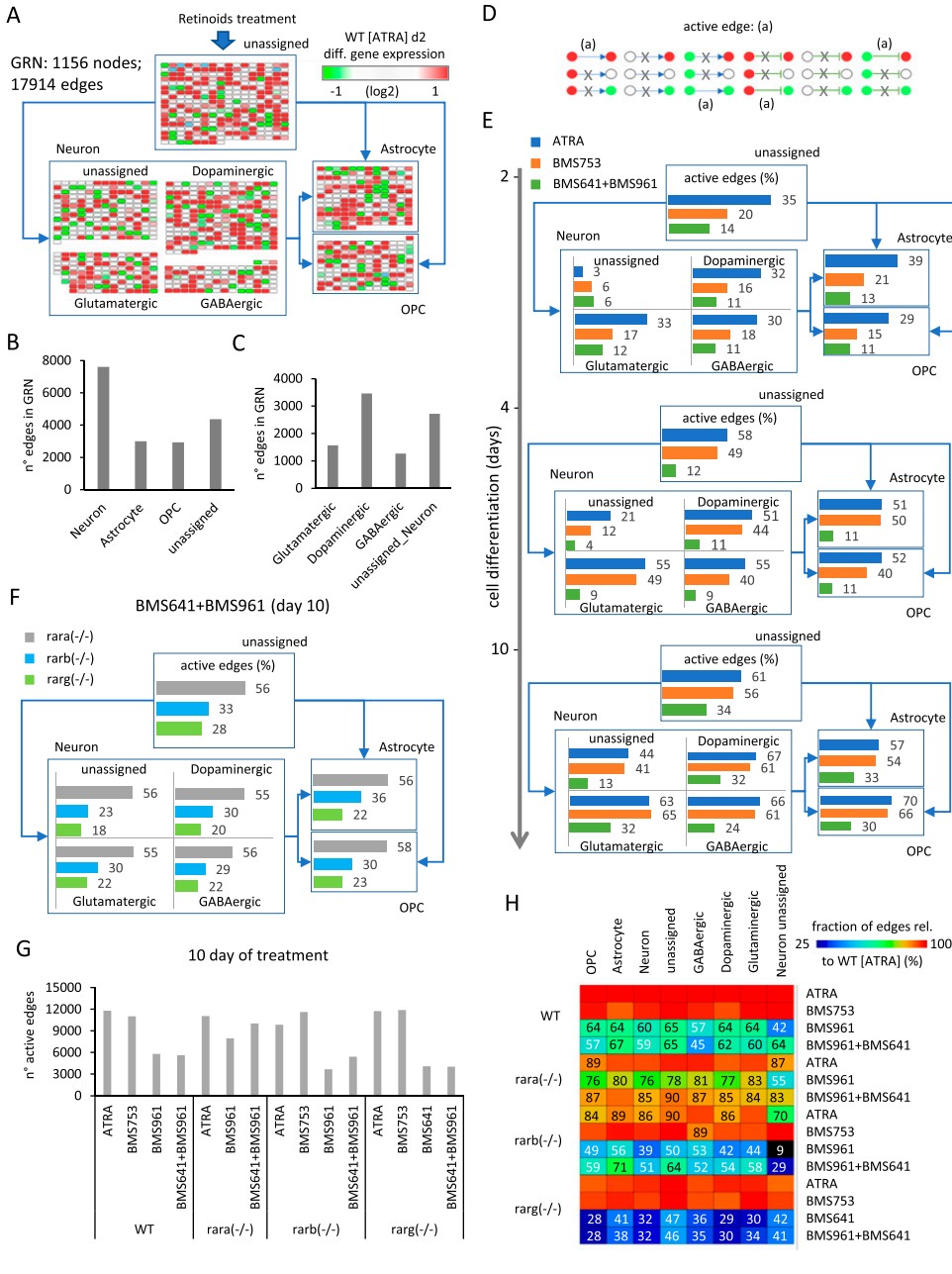

**Figure 3. Active gene regulatory wire reconstruction during cell specialization driven by retinoids.**
**(A)** Structure of the reconstructed gene regulatory network (GRN) displaying differentially expressed genes stratified into four major groups: neuronal cell markers (582 nodes), astrocytes (161 nodes), oligodendrocyte precursors (OPCs: 133 nodes), and a fourth group composed of genes not retrieved in none of the previous classifications (unassigned: 280 nodes). Nodes associated with the neuronal group have been further stratified on dopaminergic (214 nodes), glutamatergic (111 nodes), GABAergic (87 nodes), or unassigned (170 nodes) neurons. For illustration purposes, all edges were removed and replaced by simplified connectors (blue arrows). The color code associated with nodes reflects the differential gene expression levels in WT P19 cells after 2 d of all-trans retinoic acid (ATRA) treatment. **(B, C)** Number of edges interconnecting nodes retrieved on each of the aforementioned groups. **(D)** Scheme illustrating all potential types of node states (active: red; repressed: green; and unresponsive: white) and their inter-relationships defined by the illustrated edges (positive regulation: arrow connector; negative regulation: t-shaped connector). "Active edges" (a) correspond to transcriptionally relevant node/edge relationships and are conserved during the analytical processing of the GRN illustrated in (A). **(E)** Temporal transcription evolution of the reconstructed GRN during WT P19 cell differentiation. Illustrated barplots correspond to the fraction of active edges (as defined in (D)) relative to the total edges (displayed in (B, C)) issued from the treatment with either the ATRA, the RARα-specific agonist BMS753, or the combination of RARβ and RARγ agonists (BMS641 + BMS961). **(F)** Barplots corresponding to the fraction of active genes after 10 d of treatment with the RARβ and RARγ agonists (BMS641 + BMS961) of P19 Rar-null mutant lines. **(G)** Number of total active edges retrieved in GRNs issued from 10 d of treatment with the indicated retinoids and over the different P19 lines. Notice that the number of active edges on the Rara(−/−) line treated with the combination of BMS641 + BMS961 agonists leads to similar levels to those observed on the gold-standard WT line treated with the pan-agonist ATRA. **(H)** Fraction of active edges (relative to the WT line treated with ATRA) associated with markers corresponding to the classification of specialized cells retrieved in (A) relative to those observed on the gold-standard WT ATRA–treated sample. Fraction levels higher than 90% are only displayed with the heatmap color code (red).

(TF-TG relationships), was stratified based on the presence of nodes (genes) associated with a given type of specialized cell type, as described in the aforementioned collection resource (Table S1) (Hook et al, 2018; Tasic et al, 2018; Voskuhl et al, 2019) (Fig 3A). Specifically, the GRN has been first stratified into four major groups: neuronal cell markers (582 nodes; 7,611 edges), astrocytes (161 nodes; 3,007 edges), oligodendrocyte precursors (OPC: 133 nodes; 2,929 edges), and a fourth group composed of genes not retrieved in none of the previous classifications (unassigned: 280 nodes; 4,367 edges) (Fig 3A and B). Nodes associated with the neuronal group

have been further stratified on dopaminergic (214 nodes; 3,457 edges), glutamatergic (111 nodes; 1,568 edges), GABAergic (87 nodes; 1,264 edges), or unassigned (170 nodes; 2,717 edges) neurons (Fig 3A and C).

One of the major advantages of working with reconstructed GRNs is the fact that the relevance of the system can be challenged by the coherence of the interconnected players. In this case, we define an "active edge" as a set of two nodes being differentially responsive and interconnected with a transcription regulatory relationship (active or repressed regulation) coherent with the gene expression status of the interconnected nodes (e.g., active genes require to be

interconnected by active transcription regulatory edges; Fig 3D). Hence, during differentiation of WT P19 cells treated with the pan-agonist ATRA, the fraction of active edges passes from ~30% to ~50% and finally ~60% when evaluating readouts at 2, 4, and 10 d of treatment associated with specialized cell types (Fig 3E). Interestingly, WT P19 cells treated with the RARα agonist (BMS753) present a lower number of active edges after 2 d of treatment (21% for astrocytes, 15% for OPCs, and ~17% for the specialized neurons), but recovered similar levels than those observed with the ATRA treatment in the late time-points. In contrast, the use of the combination of the RARβ+γ agonist (BMS641 + BMS961) raises the levels of active edges to barely ~30% after 10 d of treatment (Fig 3E). Although this poor performance is also observed in P19 lines deficient for the RARβ or the RARγ receptor, the Rara(−/−) P19 mutant line revealed a significant gain in the number of active edges (56% for astrocytes, 58% for OPCs, and ~55% for specialized neurons) (Figs 3F and S4).

Indeed, although GRNs corresponding to WT P19 cells treated with BMS961 and BMS641 agonists present ~half of active edges observed on ATRA treatment conditions (5,628 edges), P19 Rara(−/−) mutant cells under the same retinoids' treatment lead to GRNs presenting 10,044 edges interconnecting responsive genes (Fig 3G). The comparison between the number of active edges under different conditions and that observed on the gold-standard P19 WT ATRA treatment reveals that the Rara(−/−) mutant treated with BMS961 and BMS641 agonists leads to a recovery of >80% for all cell-specialized groups (Fig 3H). In contrast, the same treatment on WT P19 cells reaches levels of ~60% for most of the groups, with the exception of the GABAergic neuronal subtypes where only 45% of edges observed on ATRA treatment are recovered.

Overall, the reconstructed GRN describing cell type specialization during retinoid-driven cell differentiation reveals the fraction of reactivated edges by the synergistic activation of the RARβ and RARγ receptors, notably in the Rara mutant line.

### Enhanced restoration of neuronal cell specialization driven by RARγ and RARβ receptors requires to bypass gene programs controlled by the RARα receptor

A deeper analysis of the reconstructed GRN revealed a twofold increase in the number of active edges for the P19 Rara(−/−) mutant relative to the WT line when treated with RARβ+γ agonists (BMS641 and BMS961). Such enhanced performance could be explained by a functional redundancy of RAR subtypes, as previously demonstrated during the early phases of endodermal (F9) and neuronal (P19) cell differentiation (Roy et al, 1995; Taneja et al, 1995, 1996). Specifically, we speculate that in the absence of the RARα receptor, the synergistic activation of the RARβ and RARγ receptors could drive the activation of RARα-specific programs. Similarly, such RARα-specific programs might remain "inhibited" in WT cells (for instance, because of the unliganded binding of the RARα receptor) despite the combined exposure to RARβ+γ agonists.

To address this hypothesis, we first identified the RARα agonist (BMS753)–specific programs, corresponding to the common active edges between WT, Rarb(−/−), and Rarg(−/−) mutant lines treated with this synthetic ligand (Fig 4A). The obtained 9,328 active edges were then intersected with those observed on WT or Rara(−/−) mutant lines treated with RARβ+γ agonists, to reveal those

programs commonly activated by both treatments (3,806 active edges), and those specifically activated by the RARα agonist (BMS753) but inhibited in the WT line despite of the combined exposure to RARβ+γ agonists (3,830 active edges).

A close look at the "common" and "inhibited" programs revealed that, despite their distinct number of edges, most of the genes composing the "common" program are also part of the "inhibited" program (414 from 452 genes), and this observation is also conserved for the involved TFs (161 shared TFs; Fig 4B). This observation suggests that gene expression for the "common" and "inhibited" programs is differentially controlled by other molecular factors in addition to TF regulation. To address this hypothesis, we have evaluated their promoter epigenetic status (defined by the repressive mark H3K27me3 and the active mark H3K4me3), their chromatin accessibility (revealed by FAIRE-sequencing assays), and their transcriptional response (revealed by the enrichment of the RNA Polymerase II) after 2 d of ATRA treatment (GSE68291 [Mendoza-Parra et al, 2016a]). This enrichment analysis (relative to those observed on EtOH vehicle treatment) revealed that genes being either specifically "inhibited" in WT (235 genes), shared between the "inhibited" and the "common" programs (414 genes), or specifically associated with the "common" programs (38 genes) are preferentially repressed at 48 h of ATRA treatment, as revealed by the enrichment of the H3K27me3 modification (Fig 4C and D).

Genes associated with the "inhibited" program are induced between the 4 and 10 d of ATRA treatment, or the synthetic RARα agonist (BMS753), but they remain unresponsive in presence of the combined exposure to RARβ+γ agonists. This being said, the combined exposure to RARβ+γ agonists leads to their gene induction on the P19 Rara(−/−) mutant line (Fig 4E).

With the aim of confirming the role of the RARα receptor on the predicted "inhibited" program, we have performed an enrichment analysis on their associated 219 FAIRE sites (Fig 4F), by comparing them with ChIP-seq binding sites collected from the public domain. Specifically, we have used a collection of more than 40,000 public mouse ChIP-seq datasets, collected as part of our NGS-QC database (https://ngsqc.org/), among which 71 ChIP-seq public profiles correspond to RXR or RAR TFs (Mendoza-Parra et al, 2013, 2016b; Blum et al, 2020). This analysis revealed that ~21% of the FAIRE sites associated with the "inhibited" program were enriched for RARα binding sites, and ~20% for pan-RXR sites, further supported by a motif analysis revealing the enrichment of the RARα primary motif (Fig 4G and H), confirming their transcriptional response driven by the RARα/RXR heterodimer.

With the aim of summarizing this information within a gene regulatory wiring, we have first assembled the FAIRE- and H3K27me3-associated "inhibited" programs into a GRN, complemented by edges issued from public ChIP-seq binding sites associated with various RARs and RXRα receptors, and with the RXRα primary motif discovery. This summarized "inhibited RXR/RAR" GRN is composed of 85 nodes and 160 edges, on which each node has been highlighted on the basis of their promoter epigenetic status (Fig 5). To further enhance the relevance of master TFs within this "inhibited" network, we have computed their master regulatory index by simulating transcription regulatory cascades over the complete reconstructed GRN (described in Fig 3A; TET-RAMER [Cholley et al, 2018]). A ranking of the TFs on the basis of their master regulatory index allowed to identify a set of 22 TFs able to

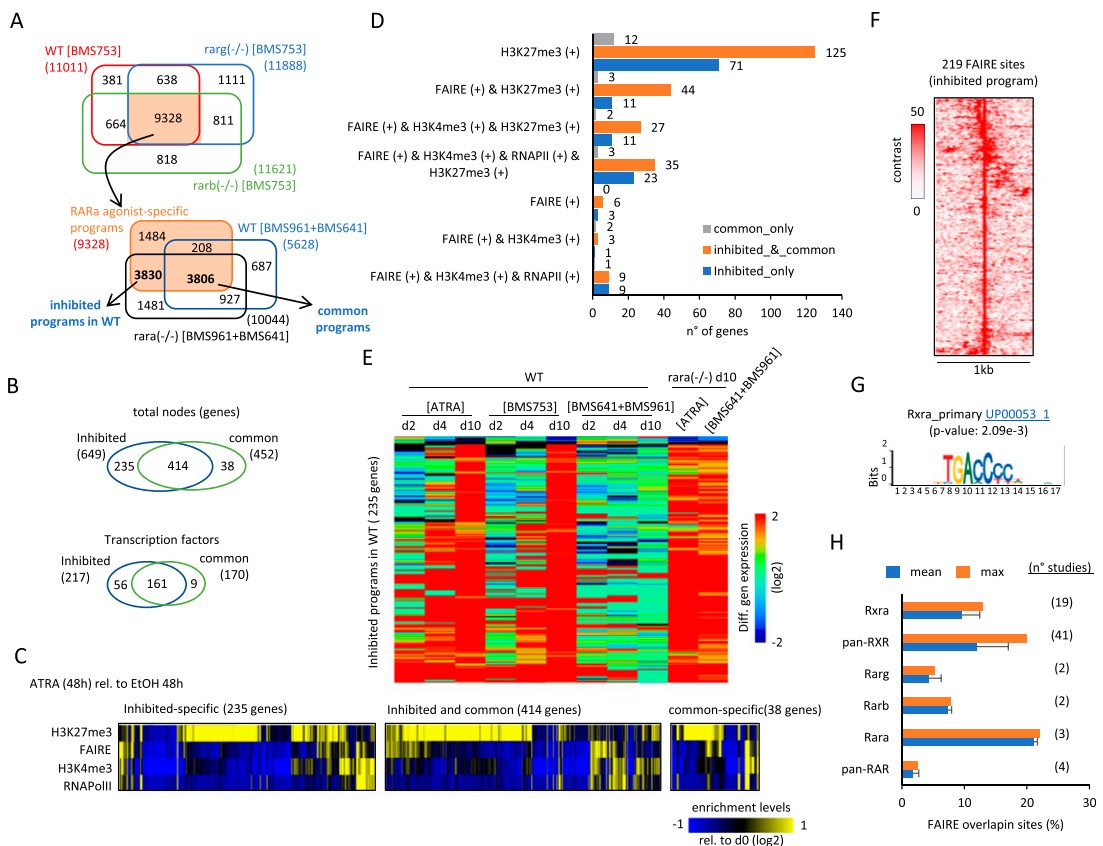

**Figure 4. Identification of a subset of active edges remained inhibited by the unliganded RARα receptor in P19 WT cells during the synergistic activation of the RARβ and RARγ.**
**(A)** Top panel: Venn diagram revealing the RARα-specific programs corresponding to the common active edges retrieved in WT, Rarg(−/−), and Rarb(−/−) P19 lines treated with the RARα agonist BMS753 (9,328 active edges). Bottom panel: Venn diagram revealing the "common programs" (3,806 active edges) driven by the RARα agonist and those responding to the synergistic activation of the RARβ + RARγ receptors; and a subset of active edges specifically driven by the RARα agonist BMS753 (3,830). This last subset is defined herein as "inhibited programs by the unliganded RARα," because they remain unresponsive on WT cells treated with the combination of RARβ + RARγ ligands (BMS641 + BMS961), but they are reactivated on the Rara(−/−) line. **(B)** Top Venn diagram: comparison between the number of genes retrieved in the "inhibited" and the "common" programs highlighted in (A). Bottom Venn diagram: comparison between the number of transcription factors retrieved in the "inhibited" and the "common" programs highlighted in (A). **(C)** Heatmap illustrating the promoter epigenetic status (repressive mark H3K27me3 and the active mark H3K4me3), the chromatin accessibility (FAIRE), and the transcriptional response (RNA Polymerase II) of genes specific to the "inhibited" or the "common" program, and those shared between these two programs after 2 d of ATRA treatment. **(D)** Number of genes presenting the indicated promoter epigenetic combinatorial status in the conditions illustrated in (C). **(E)** Heatmap displaying the differential expression levels for genes associated with the "inhibited"-specific program at different time-points and retinoid treatment. Notice that although most of these genes remained unresponsive when treated with the combination of RARβ + RARγ ligands (BMS641 + BMS961) in WT cells, they are up-regulated on the Rara(−/−) line. **(F)** Open-chromatin FAIRE sites retrieved on the promoters of the "inhibited programs by the unliganded RARα." **(G)** Motif analysis performed on the FAIRE sites presented in (F), revealing the enrichment of the RXRα primary motif. **(H)** Binding site enrichment analysis performed on the aforementioned FAIRE sites, by comparing with 71 RXR or RAR ChIP-seq publicly available profiles (NGS-QC Generator database: https://ngsqc.org/). Blue bars correspond to the mean fraction of binding sites, and orange bars correspond to the highest fraction assessed over the indicated number of studies.

regulate more than 70% of the ATRA-driven gene programs (Fig 5A and B). 15 of them present a transcriptionally active signature after 2 d of treatment, as revealed by their FAIRE-associated promoter status, whereas the remaining seven are rather repressed (H3K27me3-associated promoter status) (Fig 5C). Interestingly, this last group of TFs is composed of players like the T-box family member *Tbx18* (known to be regulated by retinoic acid during somitogenesis [Sirbu & Duester, 2006]), the transcriptional repressors *Hic1* (known to have a role in neural differentiation and tumor suppression in the central nervous system [Rood & Leprince, 2013]), *Hes6* (known to promote cortical neuronal differentiation through the repression of the TF *Hes1* [Gratton et al, 2003]), the mesoderm-specific factor *Tcf21*, the RNA binding protein *Csdc2*, the

nuclear factor IC (*Nfic*; known to regulate cell proliferation and differentiation in the central nervous system notably by modulating the expression of the miR-200b [Huang et al, 2021]) and *Zfp827* (also known as ZNF827) recently shown to negatively regulate neuronal differentiation through the expression of its circular RNA (Hollensen et al, 2020). Importantly, all these repressed factors appear interconnected within the reconstructed GRN and preferentially associated with the RARβ or RARγ receptors (Fig 5D).

Among the FAIRE-associated factors, several of them present a RXRα binding site, including the zinc-finger TF *Tshz3*—recently described as a "hub" gene involved in early cortical development (Caubit et al, 2016); *Nfix*, recently shown to drive astrocytic maturation within the developing spinal cord (Matuzelski et al, 2017); and

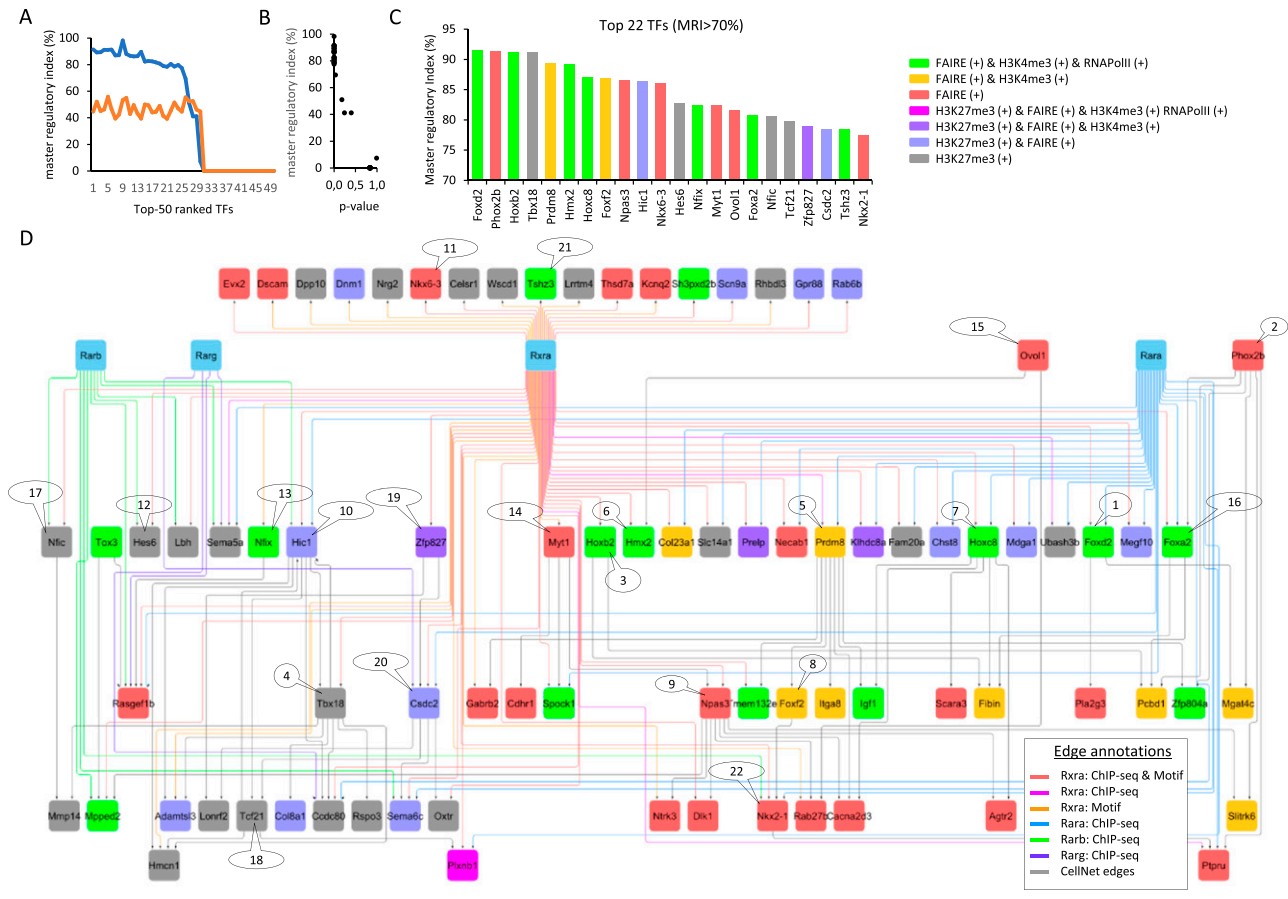

**Figure 5. Gene regulatory view of the master players inhibited by the unliganded RARα receptor during neuronal cell specialization.**
**(A)** Top 50 transcription factors retrieved within the "inhibited program" ranked on the basis of the fraction of downstream controlled genes within the reconstructed GRN (Fig 4A) (blue line). The orange line corresponds to the fraction of downstream controlled genes predicted on a randomized GRN (master regulatory index [MRI] computed as described in TETRAMER: Cholley et al, 2018). **(B)** Confidence associated with the TFs' ranking. Notice that a MRI > 70% presents the most confident *P*-values. **(C)** Transcription factors (22) presenting a MRI > 70% and colored on the basis of their promoter epigenetic combinatorial status. **(D)** Gene co-regulatory view of the 22 TFs, illustrating their most relevant (co-)regulated players and including their known relationships with RXRα and RAR nuclear receptors.

the homeodomain TFs *Hoxb2*, *Nkx6-3* (involved in the development of the central nervous system; the homeobox family factor), or *Nkx2-1*—known to control GABAergic interneurons and oligodendrocyte differentiation, and more recently described as driving astroglial production by controlling the expression of the *Gfap* (Minocha et al, 2017). Similarly, the homeobox factor *Hmx2*, the myelin transcription factor 1 (*Myt1*), or the Neuronal PAS domain protein 3 (*Npas3*) does present proximal RXRα binding sites, whereas other players like the forkhead TFs *Foxa2*, *Foxd2*, or *Foxf2*, the homeobox factor *Hoxc8*, or the histone lysine methyltransferase factor *Prdm8* do present in addition a previously described proximal RARα binding site (Fig 5D). Interestingly, *Prdm8* appears as a central player within the reconstructed GRN. Indeed, *Prdm8* controls seven other factors known to be highly expressed in nervous tissue: *Gabrb2* (the *β*2 subunit of the GABA$_A$ receptors, known to play a crucial role in neurogenesis and synaptogenesis [Barki & Xue, 2022]), the transmembrane protein *Tmem132e*, *Zfp804a* (known to regulate neurite outgrowth and involved in neuronal migration [Deans et al, 2017]), *Igf1* (insulin-like growth factor 1; synthesized by dopamine neurons [Pristerà et al, 2019]),

*Foxf2* (known to be expressed in neural crest cells leading to pericytes [Reyahi et al, 2015]), the integrin alpha 8 (*Itga8*) known to regulate the outgrowth of neurites of sensory and motor neurons, and the calcium voltage-gated channel auxiliary subunit alpha2/delta3 (*Cacna2d3*), known to be essential for proper function of glutamatergic synapses notably on the auditory brainstem (Bracic et al, 2022). As a whole, this highlighted *Prdm8* regulome appears as a critical player for controlling neuronal differentiation and specialization, in agreement with previous reports on mouse and human differentiation systems (Ross et al, 2012; Inoue et al, 2015; Cypris et al, 2020). Furthermore, our data indicate that *Prdm8* and all other factors composing the illustrated regulome in Fig 5 are driven by the RARα binding sites but can be controlled by the RARβ and RARγ receptors in the absence of the RARα receptor.

### RARγ- and RARβ-driven cell specialization programs retrieved in P19 ECs are also observed during differentiation of mESCs

To explore the relevance of the restored cell specialization capacity in P19 EC cells driven by the synergistic action of RARγ and RARβ

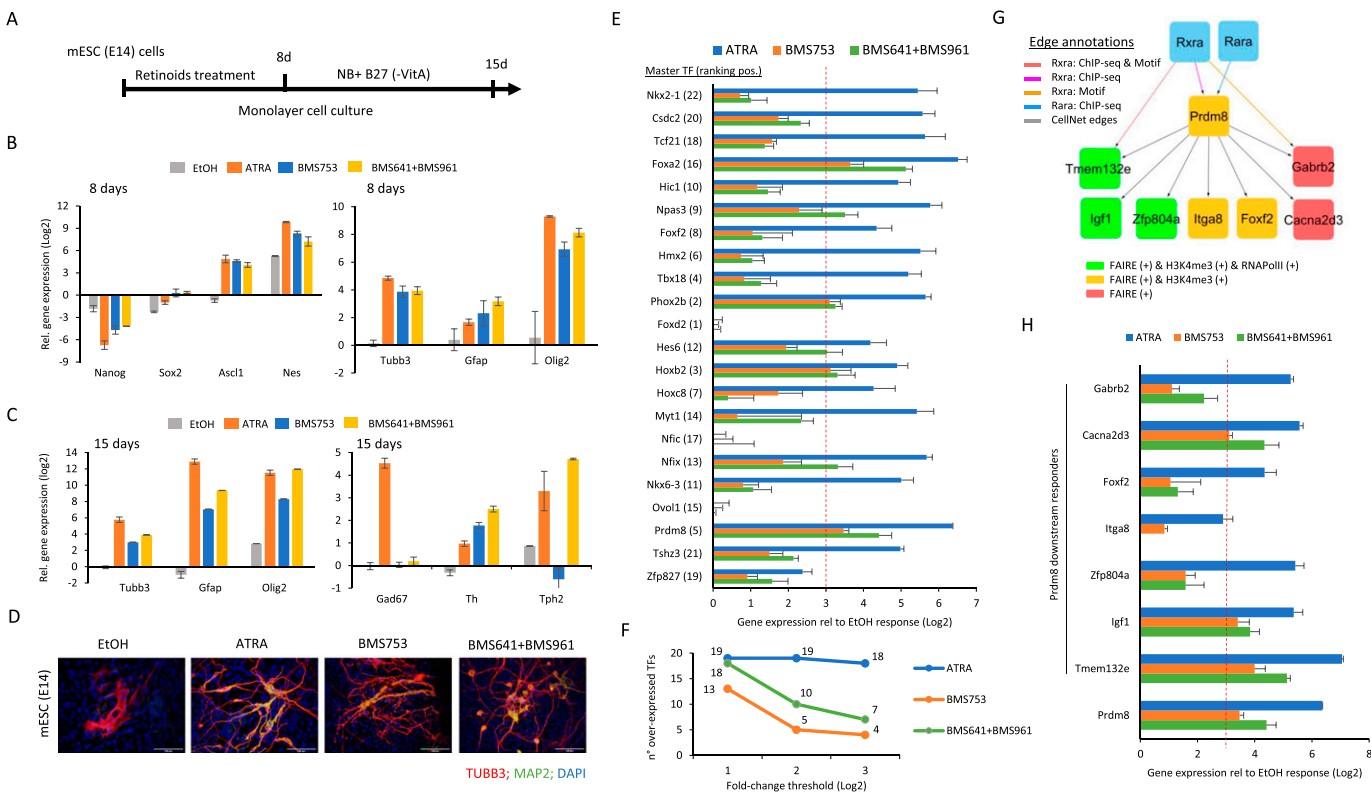

**Figure 6. RARγ- and RARβ-driven cell specialization programs retrieved in P19 ECs are also reactivated during differentiation of mouse embryonic stem cells.**
**(A)** Schematic representation of mouse ES (E14) cell differentiation assay. mES cells cultured on monolayer are exposed to retinoids during 8 d to induce cell fate commitment; then, they are cultured for seven more days on a synthetic medium (Neurobasal, NB) complemented with B27 (without vitamin A) supplements. **(B)** RT–qPCR after 8 d of differentiation, revealing the mRNA expression levels of gene markers associated with the stem cell markers (Nanog and Sox2), neuronal precursors (Ascl1, Nestin, and Tubb3), and glial cells (astrocyte-related: Gfap; oligodendrocyte-related: Olig2). **(C)** RT–qPCR after 15 d of differentiation, revealing the mRNA expression levels of gene markers associated with the neuronal marker Tubb3, the glial cell–related markers Gfap and Olig2, and the neuronal subtype markers Gad67 (GABAergic), Th (dopaminergic), and Tph2 (serotonergic). **(D)** Immunofluorescence micrograph of mES cells after 15 d of culture in the presence of either ethanol (EtOH: vehicle control), all-trans retinoic acid, the RARα agonist BMS753, or the combination of RARβ and RARγ agonists. Cells were stained for the neuronal precursor marker TUBB3 (red) and the marker for mature neurons MAP2 (green). Nuclei were stained with DAPI (blue). **(E)** RT–qPCR mRNA expression levels measured in mES cells (15 d of differentiation) corresponding to the top 22 master TFs identified in P19 cells (Fig 5). Differential gene expression is expressed relative to the expression levels observed in the presence of the ethanol control sample after 15 d of differentiation. The dashed red line demarcates a fold change threshold value of 3. **(F)** Number of overexpressed TFs under the various retinoid treatments computed at three different fold change thresholds (Log2). **(G)** Subset of the "inhibited program" retrieved in P19 cells, revealing the gene co-regulatory network associated with Prdm8. **(H)** RT–qPCR mRNA expression levels measured in mES cells (15 d of differentiation) corresponding to the different downstream targets of Prm8 revealed in P19 cells. The dashed red line demarcates a fold change threshold value of 3.

agonists, we have extended this study to the use of WT mESCs. Specifically, we have cultured mESCs in monolayer in presence of either the pan-agonist ATRA, the RARα agonist BMS753, or the combination of the RARβ-specific ligand BMS641 and the RARγ agonist BMS961. After 8 d of treatment, retinoid treatment has been replaced by Neurobasal-complemented medium for other 7 d to promote cell specialization (Fig 6A).

RT–qPCR assays performed after 8 d of mESC treatment with the aforementioned retinoids revealed the decrease in expression of stem cell markers (*Nanog* and *Sox2*)—confirming cell differentiation commitment—and the gain in the expression for neuronal precursor markers such as *Ascl1*, Nestin (*Nes*), and *Tubb3* (Fig 6B). Furthermore, gene expression induction of *Gfap* and *Olig2* was observed, indicative of the presence of glial cells in addition to neuronal commitment at this stage of differentiation. After 15 d of treatment, gene expression levels for glial cells (*Gfap* and *Olig2*) and the neuronal precursor marker *Tubb3* appeared enhanced,

and neuronal subtype markers such as *Gad67* (GABAergic), *Th* (dopaminergic), or *Tph2* (serotonergic) were also detected (Fig 6C).

Like in the case of P19 cells, in addition to the differentiation response observed in presence of ATRA or the BMS753 ligand, the synergistic treatment of mESCs with the RARβ-specific ligand BMS641 and the RARγ agonist BMS961 led to neuronal differentiation, as revealed not only by the aforementioned RT–qPCR assays, but also by immunofluorescence using the neuronal precursor marker TUBB3 and the mature neuronal marker MAP2 (Fig 6D).

Finally, to evaluate whether the synergistic action of RARγ- and RARβ-specific agonists in mESCs uses the same gene programs revealed in P19 EC cells, we have performed RT–qPCR assays targeting the gene expression of major master TFs (Fig 5). As illustrated in Fig 6E, ATRA treatment of mESCs gives rise to significant overexpression of 18 over 22 master TFs revealed in P19 EC cells (fold change [log2] ≥ 3 rel. to stem cell state; Fig 6F). The use of the RARα agonist (BMS753) leads to a strong response of at least four of the

master players (fold change [log$_2$] ≥ 3: *Prdm8*, *Hoxb2*, *Phox2b*, and *Foxa2*), and up to 13 master TFs when considering lower fold change levels (fold change [log$_2$] ≥ 1). Interestingly, the combination of RARβ+γ agonists (BMS641 + BMS961) gives rise to strong significant overexpression of the four master TFs driven by the RARα agonist (BMS753) and three other players (*Nfix*, *Hes6*, and *Npas3*), reaching up to 18 TFs activated when considering less stringent fold change levels (Fig 6E and F).

Among all responsive master players during P19 and mESC (E14) differentiation driven by the combination of RARβ+γ agonists (BMS641 + BMS961), the histone lysine methyltransferase factor *Prdm8* appeared as a major hub. Indeed, the summarized "inhibited RXR/RAR" GRN revealed in P19 EC cells up to seven factors under the direct control of *Prdm8* (Fig 6G). Interestingly, during mouse ES cell differentiation most of these downstream factors are also strongly reactivated: six of them under ATRA treatment and three of them (*Cacna2d3*, *Igf1*, and *Tmem132e*) in the presence of either the RARα agonist (BMS753) or the combination of RARβ+γ agonists (BMS641 + BMS961) (log fold change ≥ 3; Fig 6H).

Overall, the use of the combination of RARβ+γ agonists (BMS641 + BMS961) for inducing neuronal differentiation and specialization in mES (E14) cells revealed the reactivation of the same gene programs initially retrieved in P19 EC cells.

## Discussion

How cells respond to different signals to develop along defined cell lineages is a key open question to understand physiological cell differentiation, leading to the formation of not only organs and tissues, but also events such as in vitro cell reprogramming and even tumorigenesis. In this study, we specifically address the role of retinoids in activating major gene regulatory wires driving neuronal cell lineage and notably cell specialization. Previously, we have dissected the major retinoid-driven gene regulatory programs, leading to neuronal precursor formation, notably by evaluating the relevance of the activation of the RARα nuclear receptor in P19 cells with the synthetic agonist BMS753 (Mendoza-Parra et al, 2016a).

Although we have also shown in our previous study that activation of RARβ or RARγ nuclear receptors by their cognate BMS641 or BMS961 synthetic agonists is insufficient to promote neuronal differentiation, others reported that in long-term culture conditions, which included embryoid body formation, RARγ-specific ligand could induce the formation of GABAergic neurons, whereas RARα induced dopaminergic neurons (Podleśny-Drabiniok et al, 2017). In this study, we addressed neuronal differentiation in long-term culture conditions, but we have kept a monolayer culture strategy because it is known that cell–cell contact interactions retrieved on either two-dimensional (monolayer) or three-dimensional cultures could lead to different outcomes, as highlighted by the cellular complexity observed on cerebral organoid cultures (Lancaster et al, 2013). We have shown herein that activation of the RARβ receptor does not lead to mature neurons, nor to other specialized cells, whereas activation of RARγ nuclear receptors gives rise to lower yields of cell specialization than that observed when using the pan-agonist ATRA or the RARα-specific agonist BMS753. Surprisingly, their synergistic activation gave rise to

high yields of maturation, including specialized neuronal subtypes, and to other glial cells.

Previous studies demonstrated a redundancy for the activation of certain genes by distinct RXRs/RARs, and notably on Rar-null mutant lines, suggesting that in the absence of a given RAR nuclear receptor, the remaining isotypes could compensate for such dysfunction (Roy et al, 1995; Taneja et al, 1996; Chiba et al, 1997). Similarly, a synergistic 24 h activation by combining RAR isotype agonists has been attempted with P19 embryoid bodies in a recent study, suggesting that the synergistic activation of RARα and RARβ agonists might lead to TH + dopaminergic neurons, whereas RARγ and RARβ (or RARγ and RARα) might have a preference to induce Drd2+ neuronal subtypes (Podleśny-Drabiniok et al, 2017). Altogether, these studies clearly highlight the redundancy between RAR isotypes, as is further supported by Rar-null mutant experiments illustrated here. Indeed, we clearly demonstrate that Rara KO cells present an enhanced cell specialization yield relative to the WT situation. Furthermore, we have decorticated the gene programs that are inhibited by the potential action of the unliganded RARα receptor, notably by observing their activation on the Rara-null line via the synergistic action of the RARβ and RARγ agonists (BMS641 + BMS961). Among them, we have revealed the inhibition of *Prdm8*, a member of the family of histone methyltransferases, shown to play a role in the development of brain structures, notably by its capacity to regulate the transition from multipolar to bipolar morphology of cortical neurons (reviewed in Leszczyński et al [2020]).

Finally, we have expanded this study to neuronal cell specialization in mouse ES cells, notably by revealing the reactivation of the same gene regulatory programs retrieved in P19 EC cells, strongly suggesting for a general mechanism driven by the synergistic action of the RARβ and RARγ agonists (BMS641 + BMS961).

In summary, this study provides a systems biology view of the gene programs behind the previously observed redundancy between RARs, paving the way for their potential use for directing cell specialization during nervous tissue formation.

## Materials and Methods

### Cell culture

P19 cells were grown in DMEM supplemented with 1 g/l glucose, 5% FCS, and 5% delipidated FCS. P19 EC cells were cultured in a monolayer on gelatin-coated culture plates (0.1%). For cell differentiation assays, ATRA was added to plates to a final concentration of 1 μM for different exposure times. For treatment with RAR subtype–specific agonists, cells were incubated with BMS961 (RARγ-specific; 0.1 μM), BMS753 (RARα-specific; 1 μM), and/or BMS641 (RARβ-specific; 0.1 μM). After 4 d of treatment with either of the aforementioned retinoids, the medium was replaced by Neurobasal medium (ref: 21103049; Thermo Fisher Scientific) supplemented with N2 (ref: 17502048; Thermo Fisher Scientific) and B27 devoid of vitamin A (ref: 12587010; Thermo Fisher Scientific) and cultured for six more days.

Embryonic stem cells (E14) were grown in DMEM supplemented with 4.5 g/l glucose and GlutaMAX-I (ref. 11594446; Thermo Fisher Scientific), 15% FBS-ES, 5 ng/ml LIF recombinant mouse protein (ref.

15870082; Thermo Fisher Scientific), 1% penicillin–streptomycin, 1% MEM-NEAA, and 0.02% $\beta$-mercaptoethanol. E14 cells were cultured in a monolayer on poly-D-lysine–coated culture plates (0.1%). For cell differentiation assays, ATRA was added to plates to a final concentration of 1 $\mu$M. For treatment with RAR subtype–specific agonists, cells were incubated with BMS753 (RAR$\alpha$-specific; 1 $\mu$M) or BMS641 + BMS961 (RAR$\beta$ + RAR$\gamma$-specific; 0.1 $\mu$M each). After 8 d of treatment with either of the aforementioned retinoids, the medium was replaced by Neurobasal medium (ref. 11570556; Thermo Fisher Scientific) and B27 devoid of vitamin A (ref. 11500446; Thermo Fisher Scientific) and cultured for seven more days.

### Immunohistochemical staining

After 10 d of induced differentiation, cells were fixed with 4% paraformaldehyde (Electron Microscopy Sciences), followed by 3 × 5 min washes in PBS. Cells were permeabilized (Triton 0.1% in PBS; 15 min at room temperature) and blocked (10% heat-inactivated FCS in PBS for P19 cells; 0.1% Triton and 1% BSA in PBS for E14 cells during 1 h at room temperature). Cells were washed 3 × 5 min in permeabilization buffer, then incubated with the primary antibodies anti-$\beta$ III tubulin/anti-TUBB3 (ab14545; Abcam) or anti-MAP2 (ab32454). After 1 h of incubation at room temperature (for E14 cells, primary antibody incubation was done overnight at +4°C), cells were washed 3 × 10 min with permeabilization buffer followed by incubation with a secondary antibody (donkey anti-mouse IgG [H + L] antibody Alexa 555; Invitrogen A-31570; donkey anti-rabbit IgG [H + L] antibody Alexa 488; Invitrogen A-21206) and/or DAPI (D3571; Invitrogen). After 1 h at room temperature, cells were washed for 3 × 10 min in permeabilization buffer twice with Milli-Q water and finally mounted on microscope slides.

### RT–qPCR and RNA sequencing

Total RNA was extracted from P19 and E14 cells treated with either ATRA- or RAR-specific agonists, using the TRIzol RNA isolation reagent (ref: 15596026; Thermo Fisher Scientific) or RNeasy Mini Kit (ref. 74104; Qiagen). 0.5–1 $\mu$g of the extracted RNA was used for reverse transcription (HIGH CAPACITY CDNA RT; ref: 4368814; Applied Biosystems). Transcribed cDNA was diluted fivefold and used for real-time quantitative PCR (QuantiTect SYBR Green Kit; ref: 204145; Qiagen). RNA-sequencing libraries were produced with the NEBNext Ultra II RNA Library Prep Kit for Illumina (E7770). Libraries were sequenced within the French National Sequencing Center, Genoscope (150-nt pair-end sequencing; NovaSeq Illumina).

### Primary bioinformatics analyses

Fastq files were qualified with the NGS-QC Generator tool (Mendoza-Parra et al, 2013, 2016b). Reads from fastq files were mapped to the mouse mm9 reference genome using Bowtie 2.1.0 under default parameters. Mapped reads were associated with known genes with featureCounts. RNA-seq analyses were done with the DESeq2 R package. t-Distributed stochastic neighbor embedding analysis was performed with the R package Rtsne. Heatmap matrix display was generated with MeV 4.9.0. Gene ontology analyses were performed with DAVID Bioinformatics Resources (https://david.ncifcrf.gov/).

### Collection of gene markers associated with specialized cells

Gene markers associated with neurons (1,352 genes), astrocytes (501 genes), and oligodendrocyte precursors (OPCs: 501 genes) were collected from the supplementary material (Dataset_S02) of Voskuhl et al (2019). Gene markers associated with GABAergic (318 genes) and glutamatergic (311 genes) neuronal subtypes were collected from Table S9 of Tasic et al (2018). Gene markers for dopaminergic (Th+) (513 genes) neuronal subtypes were collected from Table S2 of Hook et al (2018). This assembled collection is available in our supplementary material (Table S1).

### Promoter epigenetic status analysis and RAR/RXR enrichment

P19 epigenetic readouts assessed for the repressive mark H3K27me3 and the active mark H3K4me3, the chromatin accessibility—revealed by FAIRE-sequencing profiling—and the transcriptional response (RNAPII) after 2 d of ATRA treatment were collected from our previous published study (GSE68291 [Mendoza-Parra et al, 2016a]). Normalized enrichment levels at gene promoter regions, relative to EtOH control profiles, were used for predicting their epigenetic combinatorial status. Enrichment heatmaps and mean density plots for FAIRE readouts at gene promoter regions (±500 bp) were obtained with seq-MINER 1.3.4.

FAIRE site motif analysis has been performed with the MEME Suite 5.4.1. RXR/RAR enrichment on FAIRE sites was inferred by comparing them with > 40,000 mouse ChIP-seq binding sites collected from the public domain, as part of our NGS-QC database (https://ngsqc.org/). Among all mouse ChIP-seq collected data, 71 ChIP-seq public profiles correspond to RXR or RAR TFs (Mendoza-Parra et al, 2013, 2016b; Blum et al, 2020).

### GRN reconstruction

Temporal transcriptomes issued from ATRA treatment were integrated with a collection of TF-TG relationships (CellNet [Cahan et al, 2014]) with the help of our previously developed Cytoscape App, TETRAMER (Cholley et al, 2018). TETRAMER has been also used for identifying the top 22 master TFs (master regulatory index > 70%). Gene co-regulatory wire visualization has been performed with Cytoscape 3.8.2.

## Data Availability

All RNA-sequencing datasets generated in this study have been submitted to the NCBI Gene Expression Omnibus (GEO; http://www.ncbi.nlm.nih.gov/geo/) under accession number GSE204816.

## Supplementary Information

# Acknowledgements

We thank all the members of our laboratory and the Genoscope sequencing platform for discussion. This work was supported by the institutional bodies CEA, CNRS, and Université d'Evry-Val d'Essonne. E Mathieux was supported by Genopole Thematic Incentive Actions funding (ATIGE-2017); A Koshy by the "Fondation pour la Recherche Medicale" (FRM; funding ALZ-201912009904); F Stüder by the funding 2OI9-L22 from the Institut National du Cancer (INCa); and A Bramoulle by the funding 2020-181 (INCa).

## Author Contributions

A Koshy: formal analysis, investigation, and methodology.

E Mathieux: formal analysis, investigation, and methodology.

F Stüder: data curation, software, and formal analysis.

A Bramoulle: resources and methodology.

M Lieb: resources and methodology.

BM Colombo: writing—review and editing.

H Gronemeyer: supervision, funding acquisition, and writing—review and editing.

MA Mendoza-Parra: conceptualization, formal analysis, supervision, funding acquisition, and writing—original draft.

## Conflict of Interest Statement

The authors declare that they have no conflict of interest.

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
