## [Reviewer comments · Life Science Alliance]

Life Science Alliance

Combination of RARb and RARg agonists restores neuronal maturation by hijacking RARa-driven programs

Aysis Koshy, Elodie Mathieux, François Stüder, Aude Bramoulle, Michele Lieb, Bruno Colombo, Hinrich Gronemeyer, and Marco Mendoza-Parra

DOI: <https://doi.org/10.26508/lsa.202201627>

Corresponding author(s): Marco Mendoza-Parra, *Génomique Métabolique du Genoscope*

Review Timeline:

Submission Date:	2022-07-23
Editorial Decision:	2022-08-26
Revision Received:	2022-11-10
Accepted:	2022-11-14

Transaction Report:

August 26, 2022

RE: Life Science Alliance Manuscript #LSA-2022-01627-T

Dr. Marco Antonio MENDOZA-PARRA
Institut Francois Jacob; University Evry-val-d'Essonne
2, Gaston Crémieux
Evry 91000
France

Dear Dr. Mendoza-Parra,

Thank you for submitting your manuscript entitled "Synergistic activation of RAR β and RAR γ nuclear receptors restores cell specialization during stem cell differentiation by hijacking RAR α -controlled programs". We would be happy to publish your paper in Life Science Alliance pending final revisions necessary to meet our formatting guidelines.

- you may Address Reviewer 1's request for validation in embryonic brains deficient for RAR α , RAR β or RAR γ , only if this data is readily available
- please address Reviewer 2's comments
- please upload your supplementary figures as single files and upload your table files as editable doc or excel files; please add supplementary figure legends and table legends to the main manuscript text
- please add a Running Title, Summary Blurb, keywords, and a category to our system
- please add the Twitter handle of your host institute/organization as well as your own or/and one of the authors in our system
- your GEO entry for the RNA-seq data should now be made publicly accessible

A. FINAL FILES:

B. MANUSCRIPT ORGANIZATION AND FORMATTING:

Sincerely,

Reviewer #1 (Comments to the Authors (Required)):

The authors utilized an experimental paradigm of retinoic acid-induced neural differentiation of P19 embryonic carcinoma cells and performed transcriptome, epigenetics and regulatory region analyses. Very similar methodologies were used in their previous work (Mendoza-Parra et al, Genome Res 2016), in which they dissected retinoic acid receptor subtype-specific signal pathways by means of receptor subtype-specific agonists, BMS753 for RAR α , BMS641 for RAR β , and BMS961 for RAR γ . Although the authors made knockout P19 cell lines for those receptors in the present study, novelty of this study appears quite limited. In addition, and more importantly, the in vivo relevance of the conclusions is not tested at all. Are the conclusions validated in embryonic brains deficient for RAR α , RAR β or RAR γ ?

It would be interesting to test if the same gene regulatory networks found in this study play significant roles in other experimental paradigms - for example, ES cell-derived neural precursor cell differentiation or embryonic brain-derived neural stem cell differentiation.

Reviewer #2 (Comments to the Authors (Required)):

The paper successfully analyses the effects of retinoic acid receptors synthetic agonist on stem cell differentiation into neural cell lineages. The experiments thoroughly investigate the effects of receptor stimulation under different conditions, including the concomitant use of receptors or selective exclusion. With that, the findings add to previous experiments in describing cell differentiation protocols important for research and perhaps future biomedical treatments. Despite the complexity of the topic, the manuscript was written in a clear and organized language, the research methods are adequate and the results are properly discussed.

Nevertheless, few minor text mistakes were detected:

- Page 2, Paragraph 2, first line: "Using RAR istotype knockout lines", there is a typo.
- Page 7, first paragraph: "...we generated global transcriptomes after 2, 4 and 10 days of treatment of WT P19 cells.This has been performed for samples..." These sentences need to be separated.

I consider the research of extreme relevance and recommend the approval of the manuscript after minor corrections.

Reviewer #1 (Comments to the Authors (Required)):

The authors utilized an experimental paradigm of retinoic acid-induced neural differentiation of P19 embryonic carcinoma cells and performed transcriptome, epigenetics and regulatory region analyses. Very similar methodologies were used in their previous work (Mendoza-Parra et al, Genome Res 2016), in which they dissected retinoic acid receptor subtype-specific signal pathways by means of receptor subtype-specific agonists, BMS753 for RAR α , BMS641 for RAR β , and BMS961 for RAR γ .

We acknowledge the comments of reviewer #1 and notably the reference performed to our previous work, which is indeed the seminal effort for the presented manuscript.

Although the authors made knockout P19 cell lines for those receptors in the present study, novelty of this study appears quite limited.

We consider that the presented manuscript goes beyond the fact that we have used Rar knockout P19 lines for our readouts. In fact, there are two major aspects that differ to our previous article:

- (i) In contrast to our previous article which focused on the first 72 hours of P19 cell differentiation; i.e. addressing neurogenesis, herein we present a study covering 10 days of cell culture; hence addressing cell specialization (Neuronal subtypes + glial cells)
- (ii) We have revealed for the first time that the synergistic action of the RAR β and RAR γ agonists (BMS641 + BMS961) could lead to neuronal cell differentiation and specialization.

In this context, the use of the P19 knockouts, combined with the use of the various RAR subtype-specific agonists, allowed us to decorticate the relevant gene programs that are used during this process. Hence, our manuscript does not only describe the synergistic action of the RAR β and RAR γ agonists, but in addition it reveals major players that are reactivated in absence of the RAR α receptor.

In addition, and more importantly, the *in vivo* relevance of the conclusions is not tested at all. Are the conclusions validated in embryonic brains deficient for RAR α , RAR β or RAR γ ?

While we do agree that the *in-vivo* relevance would be of interest, this aspect is out of the scope of this manuscript. The main aim of our work is to understand the major programs that are activated in presence of retinoids. Notably, the fact that the pan-agonist All-trans retinoic acid (ATRA) can activate all three Rar receptors at the same time suggests the use of Rar-specific as well as Rar redundant programs; or these programs (and their cross-talks) are not fully addressed yet by the scientific community. Our work provides a piece of information by revealing the gene programs that are reactivated by the synergistic action of the RAR β and RAR γ specific agonists.

It would be interesting to test if the same gene regulatory networks found in this study play significant roles in other experimental paradigms - for example, ES cell-derived neural precursor cell differentiation or embryonic brain-derived neural stem cell differentiation.

As part of the revised version of our manuscript, we have extended this study to mouse ES differentiation driven by retinoids action and we have demonstrated that (i) the combined treatment of mES cells with the RAR β and RAR γ specific agonists (BMS641+BMS961) leads to neuronal cell differentiation and cell specialization (glial cells + neuron subtypes detected); (ii) the synergistic action of the RAR β and RAR γ specific agonists reactivates most of the major transcription factors revealed in P19 cell differentiation and notably the Prdm8 regulome (Figure 6).

Reviewer #2 (Comments to the Authors (Required)):

The paper successfully analyses the effects of retinoic acid receptors synthetic agonist on stem cell differentiation into neural cell lineages. The experiments thoroughly investigate the effects of receptor

stimulation under different conditions, including the concomitant use of receptors or selective exclusion. With that, the findings add to previous experiments in describing cell differentiation protocols important for research and perhaps future biomedical treatments. Despite the complexity of the topic, the manuscript was written in a clear and organized language, the research methods are adequate and the results are properly discussed.

We acknowledge the fact that Reviewer #2 fully grasped the spirit of this work and that he/she finds our contribution of interest, and notably our effort to provide a clear description of our studies.

Nevertheless, few minor text mistakes were detected:

- Page 2, Paragraph 2, first line: "Using RAR istoype knockout lines", there is a typo.
- Page 7, first paragraph: "...we generated global transcriptomes after 2, 4 and 10 days of treatment of WT P19 cells.This has been performed for samples..." These sentences need to be separated.

We apologize for these typos. We have corrected them as part of this resubmitted version.

I consider the research of extreme relevance and recommend the approval of the manuscript after minor corrections.

November 14, 2022

RE: Life Science Alliance Manuscript #LSA-2022-01627-TR

Dr. Marco Antonio Mendoza-Parra
Génomique Métabolique du Genoscope
UMR8030
2, Gaston Crémieux
Evry 91000
France

Dear Dr. Mendoza-Parra,

Thank you for submitting your Research Article entitled "Combination of RARb and RARg agonists restores neuronal maturation by hijacking RARa-driven programs". It is a pleasure to let you know that your manuscript is now accepted for publication in Life Science Alliance. Congratulations on this interesting work.

DISTRIBUTION OF MATERIALS:

Again, congratulations on a very nice paper. I hope you found the review process to be constructive and are pleased with how the manuscript was handled editorially. We look forward to future exciting submissions from your lab.

Sincerely,
